# Tailoring a N-Doped Nanoporous Carbon Host for a Stable Lithium Metal Anode

**DOI:** 10.3390/nano13233007

**Published:** 2023-11-23

**Authors:** In-Hwan Lee, Yongsheng Jin, Hyeon-Sik Jang, Dongmok Whang

**Affiliations:** 1Department of Advanced Materials Science and Engineering, Sungkyunkwan University (SKKU), Suwon 16419, Republic of Korea; koggiree24@skku.edu (I.-H.L.); jinzuyeye@skku.edu (Y.J.); 2School of Semiconductor Science & Technology, Jeonbuk National University, Jeonju 54896, Republic of Korea

**Keywords:** Li metal host, metal–organic frameworks, N-doped porous carbon

## Abstract

Li metal is a promising anode candidate due to its high theoretical capacity and low electrochemical potential. However, dendrite formation and the resulting dead Li cause continuous Li consumption, which hinders its practical application. In this study, we realized N-doped nanoporous carbon for a stable Li metal host composed only of lightweight elements C and N through the simple calcination of a nitrogen-containing metal–organic framework (MOF). During the calcination process, we effectively controlled the amount of lithophilic N and the electrical conductivity of the N-doped porous carbons to optimize their performance as Li metal hosts. As a result, the N-doped porous carbon exhibited excellent electrochemical performances, including 95.8% coulombic efficiency and 91% capacity retention after 150 cycles in a full cell with an LFP cathode. The N-doped nanoporous carbon developed in this study can realize a stable Li metal host without adding lithium ion metals and metal oxides, etc., which is expected to provide an efficient approach for reliable Li metal anodes in secondary battery applications.

## 1. Introduction

In recent years, the development of mobile electronics, such as electric vehicles and drones, has increased the demand for rechargeable batteries with higher energy density [1,2,3,4]. However, graphite, the current anode material for commercial lithium ion batteries, has a low theoretical capacity (372 mAh g^−1^), which limits its ability to increase the energy density of the battery [1]. Lithium (Li) metal is a promising candidate for lithium ion battery anodes because of its high theoretical capacity (3860 mAh g^−1^) and low electrochemical potential (−3.040 V vs. SHE), which can effectively increase the energy density of the battery and overcome the limitations of graphite anodes [5,6]. However, low stability due to dendrite growth during the repeated Li metal plating process and the ‘dead Li’ formation problem during the stripping process is an obstacle to commercializing Li metal anodes [7].

To improve the stability of the Li metal anodes, various approaches have been explored, including electrolyte additives such as LiClO_4_ and LiPO_2_F_2_ [8,9], artificial solid-electrolyte interfaces (SEIs) such as LiF and Li_2_S/Li_2_Se [10], and 3D current collectors such as 3D porous Cu foil, nickel foam, and porous carbon materials [11,12]. Among them, 3D current collectors suppress dendrite formation by reducing localized current density and mitigating Li volume changes. Initial studies of these 3D current collectors have been conducted in the direction of increasing the surface area of the current collector [13]. However, these metal foam-based 3D current collectors require a sacrifice in gravimetric energy density due to the increased weight. Therefore, lightweight porous carbon-based 3D current collectors that can replace metal foils have attracted attention [14,15,16]. Although carbon-based 3D current collectors have reduced the sacrifice of energy density, there are still limitations due to the lithiophobic nature of carbon. To solve this problem, studies have been conducted to add lithiophilic materials such as ZnO and Ag within the porous carbon [17,18,19,20,21]. However, these approaches also suffer from complex fabrication processes and reduced gravimetric energy density by adding heavy metallic materials. In contrast, porous nitrogen-doped (N-doped) carbon can effectively reduce the Li nucleation overpotential while minimizing weight increase [22]. In particular, pyridinic N in porous carbon has a much larger Li binding energy than other types of N functional groups (pyrrolic N, graphitic N), making it very lithiophilic [22]. However, there is still a lack of research and experimental validation on the performance of lithium metal anodes as a function of the amount and type of nitrogen.

In this study, we demonstrated a metal-free N-doped porous carbon host of Li metal through the simple thermal calcination of metal–organic frameworks (MOFs) to realize a lithium metal anode material that can overcome the lithiophobic nature of carbon with only the light element N to minimize the sacrifice of energy density. We also systematically confirmed the effect of the amount of nitrogen contained in the nanoporous carbon and the change in the electrical conductivity of the nanoporous carbon with the calcination temperature on the performance of the 3D current collector’s function. As the calcination temperature increased, the pyridinic N contained in the porous carbon decreased, affecting the Li plating location. For electrodes with high pyridinic N functional groups and electrical conductivity, Li plating occurred inside the porous carbon, while dendrites were observed outside when the pyridinic N functional groups were insufficient. Also, when the electrical conductivity of the porous carbon matrix was insufficient due to a low calcination temperature, dendrites were formed on the outside instead of being stored inside the porous carbon, even though the porous carbon contained enough pyridinic N. This resulted in the formation of dendrites. In the case of porous carbon, having sufficient electrical conductivity and pyridinic N, the efficiency was 95.9% even after 150 cycles, and it operated stably even after 250 h under charge and discharge conditions with a current density of 0.2 mA cm^−2^ and a capacity of 0.2 mAh cm^−2^. Furthermore, a full cell test with a LiFePO_4_ (LFP) cathode showed about 91% capacity retention after 150 cycles.

## 2. Materials and Methods

### 2.1. Material Synthesis

To synthesize CZIF-8, ZIF-8 nanoparticles were prepared with a facile precipitation method using zinc acetate dihydrate (Zn(CH_3_COO)_2_∙2H_2_O, Sigma-Aldrich 99%) and 2-methylimidazole (C_4_H_6_N_2_, Sigma-Aldrich 99%) precursors. An amount of 2 g of zinc acetate dihydrate was dissolved in 50 mL of deionized water, 10 g of 2-methylimidazole was dissolved in 50 mL of deionized water, and the above solution was added with vigorous stirring. After the color of the solution turned white, the magnetic bar was removed, and the solution was kept at room temperature for 4 h. The product (ZIF-8) was obtained, followed by careful washing and drying at 80 °C for 24 h. After that, ZIF-8 was heat-treated at various temperatures (600, 800, 900, and 1000 °C) for 3 h in Ar for calcination. The resulting powders were chemically etched with 1 M HCl to remove metallic Zn and washed with deionized water. The final products were dried at 80 °C for 24 h. The products calcinated at 600, 800, 900, and 1000 °C are referred to as CZ-6, CZ-8, CZ-9, and CZ-10, respectively.

### 2.2. Material Characterization

A scanning electron microscope (FESEM, JEOL JSM-7600F, 15 kV) combined with EDS and transmission electron microscope (TEM, JEOL JEM-2100F) was employed to analyze the morphologic and elemental characteristics of the samples. The Brunauer–Emmett–Teller (BET) specific surface area was analyzed from N_2_ adsorption/desorption isotherms with a Micromeritics ASAP 2020. X-ray photoelectron spectroscopy (XPS, ESCALAB 250) was carried out with an excitation source of Al Kα (1486.6 eV) to detect the surface electronic states and chemical nature of the elements.

### 2.3. Electrical Resistance Measurement

The electrical resistance measurement was conducted after pelleting 0.5 g of CZ powders. The diameter of the pellet die was 15 mm, and the pellet was made at a pressure of 200 bar. Electrical resistance was measured with the multimeter. 

### 2.4. Electrochemical Characterization

Coin-type cells (CR2032) were used to study the electrochemical properties of the CZ-6, CZ-8, CZ-9, and CZ-10. Before cell assembly, CZ electrodes were fabricated using a slurry composed of CZ powder (80 wt%), a conducting agent (Super-P, 10 wt%), and a binder (polyvinylidene fluoride, PVDF, 10 wt%) in N-methyl-2-pyrrolidine (NMP) solution. The slurry was coated on thin Cu foil as a current collector using a conventional casting method and then dried at 80 °C overnight in a vacuum oven. The electrolyte was 1 M Lithium bis(trifluoromethanesulfonyl)im-ide (LiTFSI) in 1,3-dioxolane (DOL) and 1,2-dimethoxyethane (DME) (volume ratio = 1:1) with 1% LiNO_3_. The electrolyte was loaded with 150 μL in each cell. Before electrochemical testing, the cells were subjected to galvanostatic cycling at 0.5 mA cm^−2^ in the voltage range of 0.001 to 3.0 V versus Li/Li^+^ for three cycles. To measure the overpotential for metallization, the cells were charged to 0 V versus Li/Li^+^, and then a constant cathodic current density of 0.2 mA cm^−2^ was applied. The cycle test was performed at 0.2 mA cm^−2^ with a cutoff capacity of 0.2 mAh cm^−2^, following 2 mAh cm^−2^ of pre-deposited Li. The coulombic efficiency was obtained with the condition of 0.5 mA cm^−2^ and 1 mAh cm^−2^. Electrochemical impedance spectroscopy (EIS) for the cells was performed over a 10^6^ to 0.01 Hz frequency range. The fitting of EIS data was conducted by using Nova. Full cells were assembled using Li electrodeposited CZ as an anode and LiFePO_4_ (LFP) as a cathode. The LFP electrode was prepared using a slurry composed of LFP powder (80 wt%), a conducting agent (Super-P, 10 wt%), and a binder (PVDF, 10 wt%). The remaining fabrication process was the same as that of the CZ electrode. The anodes were electrodeposited with 5 mAh cm^−2^ of Li before the assembly of the full cells. The cycle test was performed within 2.4–4.2 V at a scan rate of 0.5 C.

## 3. Results and Discussion

ZIF-8, one of the zeolitic imidazolate frameworks consisting of Zn^2+^ ions and 2-methylimidazole linkers, was obtained as white dodecahedral crystalline particles with a size of 300–500 nm [23]. When ZIF-8 is thermally calcined, the Zn^2+^ ions become Zn metal nanoparticles, and the 2-methylimidazole linkers become electrically conductive N-doped nanoporous carbon [24]. To obtain an effective Li metal host, a sufficient N amount and high electrical conductivity are required, which are strongly affected by the calcination temperature. If the calcination temperature is too low, porous carbon that is N-rich but has low electrical conductivity is formed; conversely, if the temperature is too high, porous carbon with high electrical conductivity but that is N-poor is formed. This N-poor porous carbon is lithiophobic, meaning that there is no driving force for lithium to be stored inside, and the mass transport resistance caused by the nanoporous structure favors lithium growth on the outer surface of porous carbon particles and causes dendrite formation. If the calcination temperature is properly controlled to realize porous carbon that is N-rich and has high electrical conductivity, Li growth inside the particles can be induced because the particles can receive electrons smoothly when storing Li metal, and a sufficient amount of N can lower the Li nucleation overpotential (Figure 1).

To identify a suitable calcination temperature for N-rich porous carbon with high electrical conductivity, ZIF-8 was calcinated at various temperatures, and the electrical conductivity and N amount of the calcinated products were determined (Figure 2). When the calcination temperature was varied from 600 to 1000 °C, the particle size decreased from ~325 to ~243 nm as the temperature increased (Figure 2g), but the particle morphology was maintained without collapse (Figure 2a–d). The morphologies of CZ-6, CZ-8, CZ-9, and CZ-10 were confirmed with TEM and the porosity of CZ-8 was confirmed with BET (inset of Figure 2a–d and Appendix A). All four samples had a nanoporous structure without any Zn nanoparticles. CZ-8 showed a typical Type I isotherm with a sharp uptake at P/P_0_ < 0.05. This can be attributed to the large fraction of micropores [24]. Furthermore, a slight uptake was observed at P/P_0_ > 0.95. This might be related to the macropores between CZ-8 nanoparticles. The formation of micropores would result from the removal of organic ligands from the ZIF-8 structure during the calcination process. The specific surface area was measured to be 1284 m^2^ g^−1^, similar to previous studies [24]. After the thermal calcination, it was found that the porous carbon doped with pyridinic, pyrrolic, and graphitic N was obtained (Figure 2e). Appendix A shows the full-scan XPS spectra for the nanoporous carbons obtained with different calcination temperatures. As the temperature increases, the area of the N 1s peak in the XPS spectra decreases, indicating that the relative amount of N decreases, and the ratio of pyridinic, pyrrolic, and graphitic N changes. To compare this quantitatively, we compared the normalized ratio (Figure 2f), and the elemental analysis with SEM-EDS confirmed the at% of N in the N-doped porous carbon (Figure 2h). The normalized ratio was obtained by calculating the area of N1s peaks. As the heat treatment temperature increased, the proportion of pyridinic N, which has the most lithiophilic properties, decreased (78 to 52%), while the proportion of graphitic N increased relatively (5 to 26%). In addition, the at% of N in the porous carbon remained in the 20% range up to an annealing temperature of 800 °C and then decreased with increasing temperature until less than 1 at% of N remained at 1000 degrees Celsius. The resistivity of the porous carbon as a function of the heat treatment temperature of ZIF-8 was checked (Figure 2h). After calcinating at 600 °C, the resistivity of the resulting porous carbon was about 1600 Ωm, while at 800, 900, and 1000 °C, the resistivity values decreased rapidly to 0.14, 0.03, and 0.02 Ωm, respectively. To understand the trend of Li deposition according to the N amount and electrical conductivity, electrodes were fabricated with each sample to check the voltage change when depositing 1 mAh cm^−2^ of lithium at a current density of 0.2 mA cm^−2^ (Figure 2i). The CZ-6, CZ-8, CZ-9, and CZ-10 electrodes show initial overpotential values of 22.2, 23.4, 24.3, and 25.5 mV, respectively, which increase with the increasing calcination temperature. On the other hand, the voltage value at 1 mAh cm^−2^, where the voltage value stabilizes, shows values of 16.8, 18.2, 18.7, and 16.8 mV for CZ-6, CZ-8, CZ-9, and CZ-10 electrodes, respectively, which are not consistent with the calcination temperature. Since the overpotential value when the voltage value is stabilized is mainly determined by the mass transport resistance, it can be inferred that Li grows on the surface of the porous carbon for samples CZ-6 and CZ-10, but Li is deposited inside the porous carbon for samples CZ-8 and CZ-9. In addition, the initial overpotential excluding mass transport resistance is 5.4, 5.2, 5.6, and 8.7 mV for CZ-6, CZ-8, CZ-9, and CZ-10 electrodes, respectively, with the largest value for the CZ-10 electrode with N remaining below 1 at%, indicating the difference in lithiophilicity among the other electrodes.

To further investigate the tendency of lithium storage as a function of calcination temperature, we examined the surface of the electrodes after depositing 2 mAh cm^−2^ of lithium on each electrode (Figure 3). First, the electrode surfaces before and after lithium deposition were compared (Figure 3a–d). The thickness of all electrodes is about 42 μm, and we found that all samples have uniformly dispersed CZ and super-P without much difference. On the other hand, the surface after depositing 2 mAh cm^−2^ of lithium showed significant differences between the electrodes (Figure 3e–h). As shown in the initial overpotential graphs, the CZ-6 electrode showed dendrite growth due to the deposition of lithium from the surface instead of being stored inside the porous carbon due to its low electrical conductivity (Figure 3e). Similarly, for the CZ-10 electrode, lithium deposition was observed on the electrode surface, which is believed to be a result of the absence of lithiophilic N sites, especially pyridinic N (Figure 3h). In the optical images, lithium is also observed to have a turbid surface that does not reflect light due to the formation of mossy dendrites on the surface of the electrode (Inset of Figure 3e–h). On the other hand, for CZ-8 and CZ-9 electrodes, no lithium was observed on the surface even after depositing 2 mAh cm^−2^ of lithium (Figure 3f,g). This indicates that the lithium was stored inside the porous carbon with sufficient lithiophilic N, especially pyridinic N, and high electrical conductivity. For CZ-9, where the amount of N is relatively less compared to CZ-8, some lithium was observed on the electrode surface, indicating that it did not completely inhibit dendrite formation (inset of Figure 3g).

To evaluate the electrochemical performance of each electrode, half-cells were fabricated and evaluated (Figure 4). First, the long-term stability of the electrodes was evaluated by plating/stripping 0.2 mAh cm^−2^ of lithium at a current density of 0.2 mA cm^−2^ (Figure 4a). Of the four electrodes, the CZ-6 electrode showed the shortest lifetime. The lithium dendrites formed from the initial lithium deposition cause many side reactions, which have a critical impact on the lifetime, especially the dead Li that can be formed during the stripping process, which greatly reduces the efficiency of the electrode and causes a larger overvoltage. For the CZ-10 electrode, the test time could not exceed 100 h (50 cycles) and showed a large overvoltage value due to the lithium dendrites formed from the initial lithium deposition, just like the CZ-6 electrode. In the case of the CZ-9 electrode, although the lithium was stored inside the porous carbon, some of the lithium formed dendrites on the surface of the electrode, and the dendrite formation could not be completely suppressed, which caused the lifetime to be longer than the previous two electrodes (CZ-6 and CZ-10) but relatively shorter than the CZ-8 electrode. On the other hand, for the CZ-8 electrode, when 2 mAh cm^−2^ of lithium was deposited, the lithium was not found on the surface of the electrode and effectively suppressed the formation of dendrites, resulting in stable cell operation after about 250 h (125 cycles).

Subsequently, we evaluated the galvanostatic lithium plating/stripping cycle with a capacity of 1 mAh cm^−2^ at a current density of 0.5 mA cm^−2^ (Figure 4b). The CZ-6 electrode did not show a stable coulombic efficiency (CE) above 90% and showed a CE of about 80% up to 50 cycles, followed by a decrease in efficiency in subsequent cycles. The CZ-10 electrode showed a relatively high CE of over 90% on average, but due to the influence of dead Li formed by lithium dendrites, the CE was not stable and fluctuated greatly, and the cell stopped working after 130 cycles. For the CZ-9 electrode, it was operated for up to 130 cycles without fluctuation of CE compared to CZ-10 electrode, but after 130 cycles, it showed unstable fluctuation of CE. This shows that the lithium is stored inside the porous carbon, which is stable compared to the CZ-6 and CZ-10 electrodes, but some dendrites formed, which seems to prevent it from being driven stably over extensive cycles. On the other hand, the CZ-8 electrode showed an stable 95.8% CE with no fluctuation even after 150 cycles. The galvanostatic lithium plating/stripping cycle with same capacity at a current density of 1 mA cm^−2^ was conducted for confirming that the CZ electrode is working at a higher current density (Appendix A). The CZ-6 electrode showed the shortest cycle life and the lowest CE. It performed only 40 cycles with a CE of 46.7%. The CZ-10 electrode showed a CE of over 90% until the 57th cycle, but the CE gradually decreased and ended up showing only 21.8% at the 90th cycle. The CZ-9 electrode showed a relatively high CE of 88% at 90th cycle, but the CE rapidly went down to 21.3% at the 98th cycle. On the other hand, the CZ-8 electrode showed a CE of over 95% at the 10th cycle, and the CE slightly decreased to 89.1% over 120 cycles. At a current density of 2 mA cm^−2^, the CZ-6, 10 electrodes showed a short cycle of life and a low CE of under 30% at the 30th cycle (Appendix A). In cases of the CZ-8, 9 electrodes, a relatively high CE of 85.0 and 80.7% were observe at the 50th cycle. However, the CZ-9 electrode showed a fluctuation of the CE, probably resulting from dead Li. These results indicate that the CZ-8 also works as a stable LMA at a higher current density. Table 1 shows the electrochemical performance of previous research and this paper. Previous research shows the host materials with tremendous electrochemical performances. However, some previous research has inorganic materials, which reduce the gravimetric energy density of batteries, for the lithiophilic properties of the host, or the large surface area, and the other has to conduct the complex fabrication processes. Our research shows comparable electrochemical performance with previous research by using the simple calcination process of ZIF-8. Additionally, CZ-8 is composed of only C and N, which can maximize the gravimetric energy density of batteries.

The voltage profile of each electrode was examined to determine the performance of the electrode after the cycles (Figure 5). For the CZ-6 electrode, the continuous formation of dendrites and dead Li during the plating/stripping processes resulted in an increase in the overpotential, which was manifested by a 26.3 mV increase (38.5 to 64.8 mV) in the voltage polarization measured at 0.77 mAh cm^−2^ at the 50th cycle compared to the 5th cycle (Figure 5e). Similarly, for the CZ-9 and CZ-10 electrodes, the voltage polarization values increased by 11.6 (40.1 to 51.7 mV, 5th to 150th cycles) and 10.8 (52.9 to 63.7 mV, 5th to 50th cycles) mV, respectively (Figure 5g,h). For the CZ-6 and CZ-10 electrodes, it was difficult to measure the voltage polarization at the 150th cycle due to the end of cell life before 150 cycles or the voltage polarization became too large. However, for CZ-8 electrode, the voltage polarization at the 150th cycle was not significantly different from that at the 5th cycle and even decreased by 3.6 mV (36.2 to 32.7) (Figure 5f). This indicates that the electrode is driven stably without dendrite and dead Li formation even after 150th cycles.

The stabilities at the interface of three electrodes were studied with electrochemical impedance spectroscopy (EIS) (Figure 5i–l). Figure 5i–l demonstrates the Nyquist plots of CZ-6, 8, 9, and 10 electrodes after the 1st and 30th cycles. The high-frequency semicircle is attributed to the surface polarization resistances caused by the SEI film (R_SEI_). R_e_ represents the resistance of current collector, electrolyte, and cell. In the case of R_e_, all electrodes show similar impedance after the 1st cycle, as shown in Appendix A. After the 30th cycle, R_e_ almost did not change, indicating that there was no degradation of the current collector and depletion of the electrolyte. For the plots obtained after the 1st cycle, the resistances via the SEI films of CZ-6, 8, 9, and 10 are similarly low as 30, 37.9, 41, and 28.4 Ω. After the 30th cycle, the surface resistance of all electrodes increased, but the amount of the increase varied greatly between electrodes. In the case of the CZ-6 and 10 electrodes, a significant increase in R_SEI_ from 30 to 131 Ω and 28.4 to 105 Ω was observed, which is attributed to the thickening of the SEI. As shown in Figure 3, Li dendrite grew on the topmost surface of CZ-6 and 10 electrodes during Li plating. The Li dendrites would cause the formation of ‘dead Li’ after extensive cycling and thus increase the resistance. On the other hand, CZ-8 and 9 electrodes show a much smaller increase in R_SEI_ from 37.9 to 63 and 41 to 70 Ω, respectively. Since Li metal is stored inside the pores of the CZ-8 and 9 electrodes, SEI is also formed inside the pores of the electrodes. Unlike the SEI formed on the Li dendrite surface, the SEI formed inside the micropore is believed to be stable because it can be physically supported by nanometer-sized carbon structures. Therefore, it is speculated that additional SEI formation is inhibited and a much smaller increase in R_SEI_ is obtained. In the case of R_ct_, it is linked to the kinetics of an electrochemical reaction. After the 1st cycle, the irreversible reaction of the porous carbon matrix formed via the calcination of ZIF-8 was not completely finished and the electrodes were not stabilized, resulting in the large R_ct_ values of 120 (CZ-6), 82.6 (CZ-8), 140 (CZ-9), and 160 Ω (CZ-10) each. After the 30th cycle, the observed difference between each electrode is dramatic. In the cases of CZ-6 and 10, the R_ct_ significantly increased to 997 and 885 Ω, respectively, due to the dead Li formed during the cycle. For CZ-8 and 9, however, the R_ct_ after the 30th cycle was even lower than that after the 1st cycle with 59.8 and 61.2 Ω, repectively. This indicates that, since Li metal is stored inside the nanopore of the electrodes, dead Li formation is inhibited, which does not result in an R_ct_ increase as in CZ-6 and 10.

To confirm the suitability of CZ-8 in practical applications, a full cell with CZ-6, CZ-8, CZ-9, and CZ-10 electrodes as the anode and LFP as the cathode was fabricated and evaluated (Figure 6). The CZ-6, which performed the worst in the half-cell test, showed the most capacity fading compared to the other electrodes in the full cell (Figure 6a). The voltage profile of CZ-6 also showed a low capacity retention of 76% compared to the initial capacity after 150 cycles (Figure 6b). The CZ-9 and CZ-10 electrodes showed relatively high capacity retention compared to the CZ-6, with 83% and 79% capacity retention, respectively, but still insufficient capacity retention (Figure 6c,d). On the other hand, the CZ-8 electrode showed the best performance compared to the other electrodes, similar to the results in the half-cell test, and showed about 91% capacity retention compared to the initial capacity after 150 cycles, confirming that it is a suitable electrode for practical application (Figure 6e).

## 4. Conclusions

Electrically conductive N-doped nanoporous carbon, which is composed of only the light elements N and C, was realized via the simple calcination of ZIF-8 to implement a Li metal host with high gravimetric energy density. The amount of N, the type of functional groups, and the electrical conductivity of the porous carbon varied with the calcination temperature of ZIF-8, and CZ-8 calcinated at 800 °C showed the best performance as a Li metal host due to its high electrical conductivity, a large amount of N, and high pyridinic N ratio. The lithiophilic nature of pyridinic N and its sufficient electrical conductivity allowed lithium to be stored inside the CZ-8 electrode, which inhibited dendrite and dead Li formation. With these characteristics, the CZ-8 electrode showed a high CE of 95.8% even after 150 cycles. In addition, the CZ-8 electrode showed stable cycling capability and low voltage polarization for more than 250 h. The full cell with LFP also showed stable performance with 91% capacity retention after 150 cycles. This study demonstrated that a stable Li metal host could be realized without the addition of commonly used lithiophilic metals and metal oxides, which will pave the way for research on increasing the energy density of Li ion batteries.

## Figures and Tables

**Figure 1 nanomaterials-13-03007-f001:**
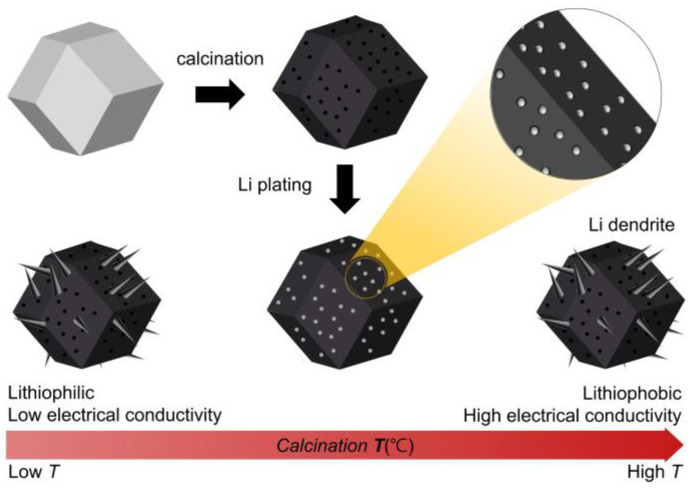
Schematic illustration of CZ fabrication process and lithium storage behavior depending on calcination temperature. The gray and black polyhedrons represent ZIF-8 and CZ, respectively. The gray sphere and cone represent Li metals.

**Figure 2 nanomaterials-13-03007-f002:**
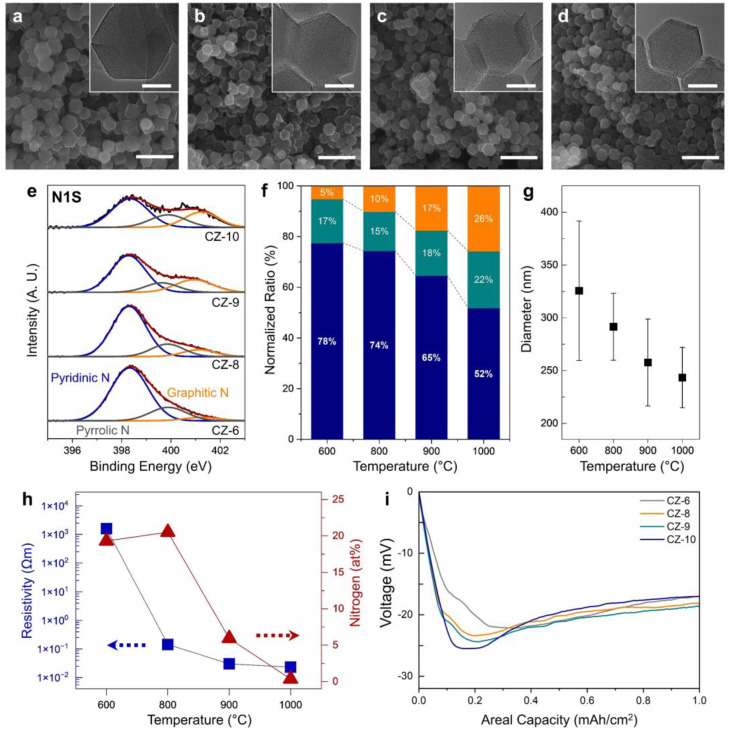
Structural characterization of CZ-6, 8, 9, 10. SEM image of CZ nanoparticles calcined at (**a**) 600 °C, (**b**) 800 °C, (**c**) 900 °C, (**d**) 1000 °C. Inset of (**a**–**d**) shows TEM image of CZ-6, 8, 9, 10 nanoparticles. (**e**) Normalized N1s XPS spectra of CZ calcined at various temperatures. (**f**) Normalized ratio of pyridinic, pyrrolic, and graphitic nitrogen of CZ-6, 8, 9, 10. (**g**) Size distribution of CZ-6, 8, 9, 10 particles. (**h**) Resistivity and nitrogen content of CZ-6, 8, 9, 10. (**i**) Voltage profiles during initial Li plating on the different electrodes at 0.2 mA cm^−2^. Scale bar 1 μm (**a**–**d**), 100 nm (inset of (**a**–**d**)).

**Figure 3 nanomaterials-13-03007-f003:**
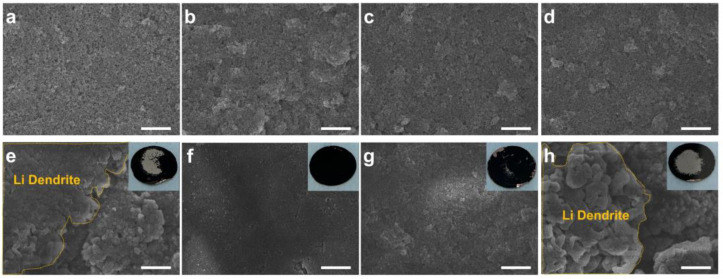
SEM image of CZ electrodes (**a**–**d**) before and (**e**–**h**) after 2 mAh cm^−2^ of Li plating. (**a**,**e**) CZ-6, (**b**,**f**) CZ-8, (**c**,**g**) CZ-9, (**d**,**h**) CZ-10. Inset of (**e**–**h**) indicates optical image of the electrodes after 2 mAh cm^−2^ of Li plating. Scale bar 10 μm.

**Figure 4 nanomaterials-13-03007-f004:**
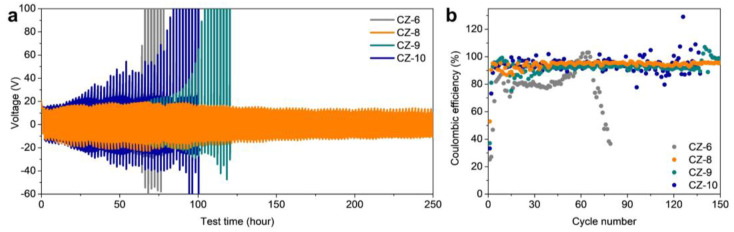
(**a**) The cycling stability of the electrodes with a fixed amount of Li plating/stripping (0.2 mAh cm^−2^) at 0.2 mA cm^−2^. (**b**) Coulombic efficiency of CZ-6, 8, 9, and 10 electrodes.

**Figure 5 nanomaterials-13-03007-f005:**
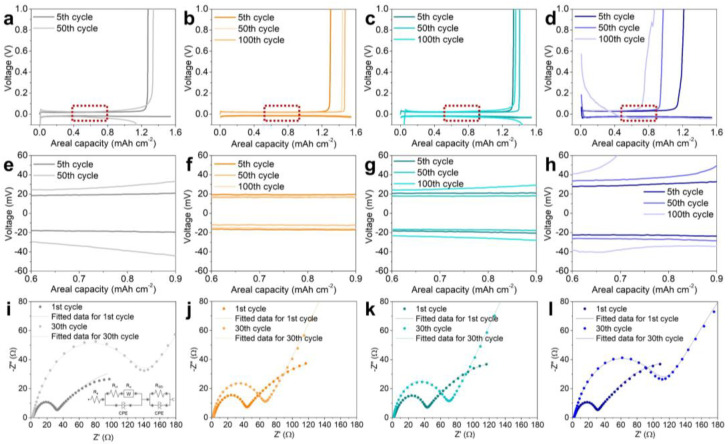
Voltage profiles of (**a**,**e**) CZ-6, (**b**,**f**) CZ-8, (**c**,**g**) CZ-9, and (**d**,**h**) CZ-10 electrodes. (**e**–**h**) The specific range of profiles at 0.6–0.8 mAh cm^−2^ is magnified. Nyquist plots after 1 and 30 cycles for (**i**) CZ-6, (**j**) CZ-8, (**k**) CZ-9, and (**l**) CZ-10 electrodes. The inset in (**j**) shows the equivalent circuit model of the electrodes.

**Figure 6 nanomaterials-13-03007-f006:**
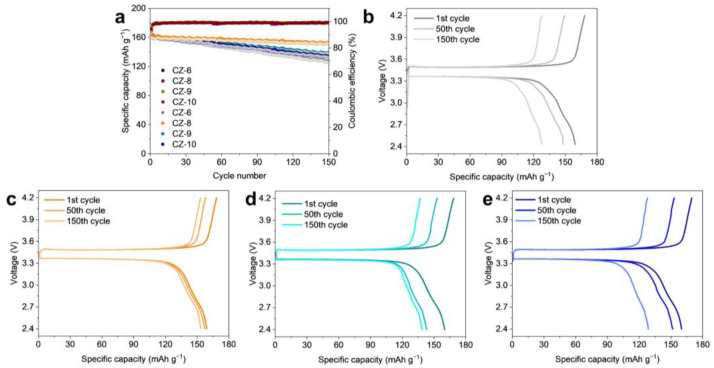
Electrochemical performance of full cell with four different electrodes; CZ-6, 8, 9, and 10. (**a**) Capacity retention and charge–discharge profiles of the electrodes. Voltage profiles of full cells with (**b**) CZ-6, (**c**) CZ-8, (**d**) CZ-9, and (**e**) CZ-10. The C rate is 0.5 C.

**Table 1 nanomaterials-13-03007-t001:** Electrochemical performances of previous research and this work.

No.	Coulombic Efficiency	Host Materials	Lithiophilic Component	Electrolyte	Ref.
1	93% (170 cycles)	N-doped porous core with Co nanoparticles implanted into 3D carbon nanotubes	N-doped carbon, Co nanoparticles	1 M LiTFSI in DOL/DME (1:1 *v*/*v*) with 0.5 M LiNO_3_	[20]
2	97.5% (80 cycles)	Wrinkled graphene cage	Au	1 M LiPF_6_ in EC/DEC with 10% FEC and 1% VC	[21]
3	95% (200 cycles)	Mesoporous carbon	-	1 M LiPF_6_ in EMC:FEC = 7:3	[16]
4	97% (250 cycles)	3D porous copper	-	1 M LiTFSI in DOL/DME (1:1 *v*/*v*) with 1% LiNO_3_	[11]
5	97% (50 cycles)	3D porous copper	-	1 M LiTFSI in DOL/DME (1:1 *v*/*v*)	[13]
6	97% (200 cycles)	Carbon nanofiber@reduced graphene oxide nanosheet	N-doped carbon	1 M LiTFSI in DOL/DME (1:1 *v*/*v*) with 2 wt% LiNO_3_	[14]
7	98% (600 cycles)	N-doped hollow porous bowl-like hard carbon/reduced graphene nanosheet	N-doped carbon	1 M LiTFSI in DOL/DME (1:1 *v*/*v*) with 0.2 M LiNO_3_	[15]
8	96% (340 cycles)	Argentohile embedded in the 3D carbon scaffold	Ag	1 M LiTFSI in DOL/DME (1:1 *v*/*v*) with 1 wt% LiNO_3_	[18]
9	95.8% (150 cycles)	N-doped nanoporous carbon		1 M LiTFSI in DOL/DME (1:1 *v*/*v*) with 1% LiNO_3_	This work

## Data Availability

Data underlying the results presented in this paper are not publicly available at this time but may be obtained from the authors upon reasonable request.

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
