# Peer review of "Tailoring a N-Doped Nanoporous Carbon Host for a Stable Lithium Metal Anode"

_nanomaterials, 2023, doi:10.3390/nano13233007_

Round 1
Reviewer 1 Report
Comments and Suggestions for Authors
Lee et al studied N-Doped nanoporous carbon host for stable lithium metal anode. The work shows interesting results and can be considered for acceptance after proper revisions.
1. Since the name of the journal is Nanomaterials, the authors should provide more characterization of the material on the nanoscale. In view of this, TEM, HRTEM, SAED and other results should be provided to meet the requirements of the journal.
2. The authors pointed out that the as-synthesized material was Nanoporous Carbon, but the characterization of the reaction material with nanoporous morphology and key evidence were not provided. It is suggested that the authors provide TEM and BET data and analyze the nanoporous structure.
3. A table should be used to compare the electrochemical performance of this paper and published works.
4. The loading of the electrolyte should be provided.
5. Clear innovations should be refined and stated.
6. Introduction. More Li metal anodes should be described and referenced. The following literature is for your reference.
[1] Rare Metals 2023, doi:10.1007/s12598-023-02360-7
[2] Batteries 2023, 9, 391. doi:10.3390/batteries9080391
[3] Battery Energy 2022, 1 ,20210012. doi:10.1002/BTE2.20210012
Comments on the Quality of English LanguageMinor editing of English language required.
Author Response
We sincerely appreciate you for the valuable comments on our manuscript. Please see the attachment.

Reviewer 2 Report
Comments and Suggestions for Authors
Authors synthesized a metal-free N-doped porous carbon host of Li metal anode. While there are various kinds of relevant materials published for Li-metal anode, especially the electrochemical performances obtained for this material during Li platting and stripping process are below the standard. Authors are suggested to enrich the manuscript with additional results.
1. It is recommended to provide the electrochemical performances at different current densities such as 0.1, 0.5, 1 and 2 mA cm-2 to explain the role of the metal-free N-doped porous carbon host of Li metal anode during the cycling process (EcoMat, 2020, 3).
2. XPS spectra in Figure 2e and 2f shows the difference in pyridinic, pyrrolic and grapahtic N. However, the grapritic N peaks in Figure2e (low intensity peak) is not consistent with figure 2f (higher Graphtic contect) which might be due to peak fitting or exerimental errors, Authors should double check and redrwa the the N XPS peak fitting.
3. There are four different electrodes CZ-6, CZ-8, CZ-9, and CZ-10. The CZ-8 shows high cycling performance. Authors should provide EIS analysis to elaborate the importance of optimized (CZ-8) composites (Adv. Mater., 2020, 32, 2002170).
Comments on the Quality of English Language
NA
Author Response

(The authors gave the same response as above.)

Reviewer 3 Report
Comments and Suggestions for Authors
I have reviewed the article entitled “Tailoring N-Doped Nanoporous Carbon Host for Stable Lithium Metal Anode” by In-Hwan Lee et al submitted to Nanomaterials / MDPI. Owing to the ultrahigh theoretical specific capacity (3861 mAh g−1) and low redox potential (−3.04 V versus standard hydrogen electrode), metallic lithium has been regarded as one of the most promising anode materials for high-energy-density rechargeable batteries. Therefore, metallic lithium has been considered as an ideal anode candidate for future high energy density lithium batteries, however, it has severe safety issues and the formation of SEI layer led to low coulombic efficiency and poor cycle performance – how the authors have addressed these issues? Please justify.
The authors have reported the synthetic strategy while controlling the amount of lithophilic N and N-doped porous carbons to optimize the performance metrics. Have these porous buffered the compressive stress during long-term cycling? Is the role of lithiophilic nitrogen-doped carbon sites to homogenize Li deposition and guide the crystal nucleus distribution, which is unclear in the current version of the manuscript? Please explain.
While revising the manuscript, in addition to the above issues, the following items must be incorporated.
1. To mitigate the problem faced with metallic lithium anode, Minakshi et al have reported an aqueous lithium secondary battery system. The key papers such as doi.org/10.1016/j.jallcom.2011.03.044; and doi.org/10.1016/j.jpowsour.2005.03.184has stated without any SEI and dendrite formation. Please include and discuss.
2. The number of cycles to 180 cycles is limited.
3. Is there any data for different C-rates?
4. Include the nucleation overpotential and polarization values for lithiophilic nitrogen-doped samples/
5. The rate capability and stale cycle life must be addressed.
6. The functional lithiophilic nitrogen-doped carbon sites and their role must be stated.
Comments on the Quality of English LanguageSome minor improvements in language and overall polishing will help.
Author Response

(The authors gave the same response as above.)

Round 2
Reviewer 1 Report
Comments and Suggestions for Authors
The authors have carefully revised the paper as suggested by the reviewers, in which case the paper can be considered for acceptance.
Reviewer 2 Report
Comments and Suggestions for Authors
The authors made an effort to address the comments, but due to the limited one-week revision period, they couldn't do so effectively. I recommend that the authors allocate more time to address the comments, with special attention to the EIS measurements, which are crucial for comprehending Z8's performance. Providing experimental evidence, such as EIS results, is essential to back up the optimized performance of CZ8, as charge transfer resistance is a critical aspect in electrodes.
Furthermore, please ensure that comment no. 1 and comment no. 3 are addressed in the initial revision before submitting the revised version.
Comments on the Quality of English LanguageNA
Reviewer 3 Report
Comments and Suggestions for Authors
This reviewer went through the revised parts of the manuscript and the responses made by the authors. Overall, it is satisfactory. The revised version is suitable for publication.
